# OpenReview forum: "SIPO: Stabilized and Improved Preference Optimization for Aligning Diffusion Models"
_ICML.cc/2026/Conference — ICML 2026 regular_

### Official Review · Reviewer_D1SB · 2026-03-08

**Soundness:** 2
**Presentation:** 3
**Significance:** 3
**Originality:** 2
**Overall Recommendation:** 5
**Confidence:** 3

**Summary:**

This paper addresses a practical and crucial issue in preference alignment within diffusion models: existing DPO-based methods in diffusion/flow-based generation scenarios often suffer from training instability, parameter sensitivity, late-stage performance degradation, and even collapse due to differences in gradient variance across different timesteps and mismatches between offline preference data and the current policy distribution. The authors conduct a systematic analysis of this problem, pointing out that early timesteps often correspond to lower importance weights, easily introducing noisy gradients, while mid-to-late timesteps are more effective for preference optimization. Based on this, the paper proposes the SIPO (Stabilized and Improved Preference Optimization) framework. The first component is DPO-C&M, which uses timestep-aware clipping and masking to suppress updates from unreliable timesteps. The second component adds timestep-wise clipped importance re-weighting to alleviate off-policy mismatch and better emphasize informative updates. The authors conducted experiments on models such as SD1.5, SDXL, CogVideoX-2B/5B, and WanX-1.3B, covering both text-to-image and text-to-video tasks. The results show that SIPO outperforms baselines such as Diffusion-DPO in terms of stability, hyperparameter robustness, and multiple automatic metrics and human evaluations.

**Compliance With Llm Reviewing Policy:**

Affirmed.

**Final Justification:**

Thank the authors for the detail reply. All my concerns have been solved, I will directly raise my rating.

One interesting thing is the authors claim to establish a "soft upper bound", specifically, a controlled bias designed to prevent exploding variance. This method aligns with the training strategy adopted in recent work [a], so I think it is reasonable and very sound. So now I‘m inclined to accept this paper. Good luck.

[a] Not All Directions Matter: Toward Structured and Task-Aware Low-Rank Adaptation. arXiv 2026.

**Key Questions For Authors:**

Please see the weakness part.

**Limitations:**

Yes

**Strengths And Weaknesses:**

# Strengths

One significant strength of this paper is its intriguing problem definition. Many current preference optimization efforts emphasize reward alignment or metric improvement, but discussions on "why training is unstable" and "what role different timesteps play" in diffusion-based preference learning are insufficient. This paper explicitly focuses on two points: timestep-dependent instability and off-policy mismatch. I particularly appreciate this interesting observation by the authors.

Another strength is the clear causal chain in the method design. The authors first observe that early timesteps often correspond to lower importance weights, and these timesteps are more prone to introducing high variance and unreliable updates. They then propose DPO-C&M using clipping and masking to suppress these noisy regions. Furthermore, considering the distributional drift between offline preference data and the current policy, they propose SIPO's timestep-wise importance re-weighting to mitigate off-policy bias.

# Weakness

While the theoretical section is quite lengthy, its rigor and necessity could be further strengthened. The authors describe SIPO as importance-sampled preference optimization and provide derivations in the form of shaped target distribution and KL equations, which is formally complete; however, from the paper's current presentation, this section seems more like providing a reasonable explanation for the method than offering truly compelling new theoretical conclusions. For example, the specific estimation method of importance weight, its variance control effect, and the trade-off between bias and benefit after clipping are not analyzed in depth. In other words, while there is considerable theoretical description, the key arguments that truly convince people why this method is stable and superior to existing methods still rely more on empirical results than rigorous analysis.

The method's novelty is somewhat incremental. From a component perspective, the concepts of clipping, masking, importance weighting, and off-policy correction are not new; similar ideas already exist in RLHF, offline RL, and even the DPO extension of language models. This paper's contribution lies more in adapting these mechanisms to the timestep dimension of diffusion preference optimization and providing a relatively systematic empirical integration. Therefore, I acknowledge its engineering and experiential value, but from the perspective of methodological originality, it is more like a solid improvement framework than a groundbreaking new paradigm. This is also why I'm on the borderline.

---

> ### Author Rebuttal · Authors · 2026-03-31
>
> We sincerely thank the reviewer for this thoughtful assessment. We agree that the theoretical contribution and practical significance of our work can be presented more clearly. Following your valuable suggestion, we plan to include some theoretical derivations in the main paper to better illustrate our theoretical motivation and contribution. We also highly appreciate the reviewer's professional insight regarding the "bias-variance trade-off," which is indeed a critical theoretical core. In particular, we agree that our work does not aim to re-invent clipping, masking, importance weighting, or off-policy correction as standalone concepts, since these are already well established in RL-related literature.
>
> However, we would like to clarify that our contribution is **not merely an empirical combination of known components**. The key novelty is to **rigorously connect these mechanisms to diffusion preference optimization**, forming a rigorous closed loop where **theory and experiment mutually corroborate each other**:
>
> * **Diffusion-Specific Diagnosis & Formulation:** Concretely, **Sec. 3.1** identifies the two diffusion-specific sources of instability—*low-importance early timesteps* and *off-policy drift*—rather than only reporting instability empirically. Building on this diagnosis, **Sec. 3.2** rigorously derives DPO-C&M and SIPO from an importance-sampled preference optimization formulation. This yields the principled shaped-target and KL forms, instead of introducing a purely heuristic objective.
>
> * **Theoretical Depth on the Weighted Objective:** Directly addressing the variance control and bias-variance trade-off mentioned by the reviewer, **Fig. 5** and **Appendix B.2** further analyze the weighted objective itself. We mathematically demonstrate why the $q/p$-weighted form is substantially more stable. This analysis proves how the mechanism provides adaptive step sizes, establishes a soft upper bound (a controlled bias to prevent exploding variance), improves robustness to biased/off-policy samples, and ensures better numerical stability.
>
> * **Mutual Corroboration of Theory and Experiment:** Crucially, our theoretical analysis is not an isolated afterthought. The variance-bounding mechanisms and stability guarantees derived in our theory perfectly predict and explain the unprecedented empirical stability observed during long-horizon training (**Fig. 7**) and within high-variance video generation tasks (**Table 1**). This strict mutual corroboration proves that SIPO's empirical success is fundamentally rooted in its mathematical design.
>
> To sum up, the novelty of SIPO lies not in these single component alone, but in the *diffusion-specific diagnosis*, the *timestep-aware objective design*, and the *strong mutual corroboration* between our principled theory and empirical breakthroughs. Combining these components together, our SIPO offers a principled importance-weighted optimization framework that stabilizes training and mitigates off-policy mismatch by reweighting gradients according to sample likelihood and timestep quality.
>
> On the other hand, extensive results on various baseline models across both image and video generation tasks consistently reflect the superioty of our proposed SIPO compared with existing alternatives. More importantly, SIPO remains robust under online and iterative training, offering a scalable and principled approach to aligning diffusion-based generation with
> human preferences.
>
> Once again, we sincerely thank the reviewer for their valuable time and constructive feedback. Please let us know if you have any further questions.

---

> > ### Author Rebuttal · Reviewer_D1SB · 2026-04-02
> >
> > Thank the authors for the detail reply. All my concerns have been solved, I will directly raise my rating.
> >
> > After reviewing the authors' rebuttal, and Appendix B.2, I notice an interesting point: the authors claim to establish a "soft upper bound", specifically, a controlled bias designed to prevent exploding variance. This method aligns with the training strategy adopted in recent work [a], so I think it is reasonable and very sound. So now I‘m inclined to accept this paper. Good luck.
> >
> > [a] Not All Directions Matter: Toward Structured and Task-Aware Low-Rank Adaptation. arXiv 2026.

---

> > > ### Author Response · Authors · 2026-04-02
> > >
> > > Dear Reviewer,
> > >
> > > We are very pleased that our rebuttal has addressed your concerns, and we sincerely appreciate your time, efforts, and constructive suggestions, which have helped us improve our paper significantly.  We will incorporate the contents of our rebuttal into the revised manuscript to better present our theoretical and empirical contributions.
> > >
> > > We also thank you for pointing out the relevant concurrent work. The referenced study offers useful perspectives that are highly complementary to our work. Please let us know if you have any further questions.

---

### Official Review · Reviewer_PGBM · 2026-03-10

**Soundness:** 3
**Presentation:** 3
**Significance:** 3
**Originality:** 3
**Overall Recommendation:** 4
**Confidence:** 4

**Summary:**

This paper addresses two fundamental challenges: training instability and off-policy bias in aligning diffusion models with human preferences via Stabilized and Improved Preference Optimization (SIPO). To address these issues, SIPO introduces clipping and masking uninformative timesteps to filter out low-quality updates, and combines a a timestep aware importance re-weighting paradigm to fully correct off-policy bias and emphasize informative updates throughout the alignment process.

**Compliance With Llm Reviewing Policy:**

Affirmed.

**Final Justification:**

The author answered my questions carefully.

**Key Questions For Authors:**

see the weakness

**Limitations:**

yes

**Strengths And Weaknesses:**

**Strengths**

1.The training process is greatly stabilized and parameter robust: SIPO effectively solves the training instability problem caused by high variance in the early stages by clipping and masking early time steps with less information through the DPO-C&M mechanism.

2.Effective correction of off-policy bias: This method introduces an importance re-weighting mechanism with time-step awareness, which can adaptively reduce the bias caused by the inconsistency between the distribution of offline data and the current policy model.

**Weaknesses**

1.There is a lack of comparisons with methods similar to Diffusion-DPO, such as SPO and KTO.

2. LLM Applicability Claimed but Not Demonstrated. Section B.3 claims SIPO is modality-agnostic and applicable to LLMs, but no LLM experiments are included in the main paper.

3.There are too few visual comparisons, and the main paper lacks visualization results for the video dataset.

4.Section E.5 compares HPSv2 with varying clipping parameters ε on only two models (SD1.5, SDXL), lacking analysis on video and flow-matching models.

---

> ### Author Rebuttal · Authors · 2026-03-31
>
> > **Weaknesses 1**: There is a lack of comparisons with methods similar to Diffusion-DPO, such as SPO and KTO.
>
> We thank the reviewer for the suggestion. Following the reviewer’s suggestion, we additionally compare against **SPO** and **Diffusion-KTO** under the same **DyMO** evaluation protocol for SD1.5-based methods. We evaluate on the **Pick-a-Pic validation set (500 prompts)** and report **PickScore**, **HPSv2**, **Aesthetic Predictor**, and **ImageReward**. As shown below, **SIPO** achieves the best **PickScore (21.78)**, **HPSv2 (27.83)**, and **Aesthetic (5.911)** among the compared methods. Relative to Diffusion-KTO, our method achieves higher PickScore, HPSv2, and Aesthetic, while obtaining a slightly lower but comparable **ImageReward**.
>
> | Method        | PickScore ↑ |   HPSv2 ↑ | Aesthetic ↑ | ImageReward ↑ |
> | ------------- | ----------: | --------: | ----------: | ------------: |
> | SD1.5         |       20.73 |     23.41 |       5.337 |        0.1697 |
> | Diffusion-DPO |       20.97 |     26.56 |       5.594 |        0.2989 |
> | SPO           |       21.46 |     26.71 |       5.702 |        0.2321 |
> | Diffusion-KTO |       21.15 |     27.19 |       5.697 |        **0.6156** |
> | SIPO (ours)   |   **21.78** | **27.83** |   **5.811** |    0.5964 |
>
> > **Weaknesses 2**: LLM Applicability Claimed but Not Demonstrated.
>
> Thank you for pointing this out. We agree that our wording in Section B.3 was not clear enough. What we intended to express is that the objective formulation is modality-agnostic and can in principle be applied to LLM preference alignment. To support this, we also include **appendix LLM results**. Following SimPO, we use **Mistral-7B-Instruct** and **Llama-3-8B-Instruct**, construct preference pairs from **UltraFeedback** by sampling 5 responses per prompt from the SFT model, and use **PairRM** to select the preferred/dispreferred responses. We evaluate on AlpacaEval 2 and MT-Bench. As shown below, **SIPO** consistently improves over SFT and achieves the best WR on both backbones. We will revise this part to avoid overclaiming and present the LLM result more clearly as appendix evidence of extensibility.
>
> | Method   | Mistral LC (%) | Mistral WR (%) | Mistral MT-Bench | Llama-3 LC (%) | Llama-3 WR (%) | Llama-3 MT-Bench |
> | -------- | -------------: | -------------: | ---------------: | -------------: | -------------: | ---------------: |
> | DPO      |           26.8 |           24.9 |              7.6 |           40.3 |           37.9 |              8.0 |
> | IPO      |           20.3 |           20.3 |              7.8 |           35.6 |           35.6 |          **8.3** |
> | KTO      |           24.5 |           23.6 |              7.7 |           33.1 |           31.8 |              8.2 |
> | R-DPO    |           27.3 |           24.5 |              7.5 |           41.1 |           37.8 |              8.0 |
> | SimPO    |       **32.1** |           34.8 |              7.6 |       **44.7** |           40.5 |              8.0 |
> | **SIPO** |           31.8 |       **35.1** |          **7.9** |           43.5 |       **41.6** |              8.2 |
>
> >**Weaknesses 3**: There are too few visual comparisons, and the main paper lacks visualization results for the video dataset.
>
> Thank you for this suggestion. We agree that the visual comparison in the current main paper is limited. In fact, we do provide additional video-generation cases in the appendix. We will make this clearer in the revision and also move/add more representative video visualization examples to the main paper, so that the qualitative comparison is better aligned with the quantitative results.
>
> >**Weaknesses 4**: Section E.5 compares HPSv2 with varying clipping parameters ε on only two models (SD1.5, SDXL), lacking analysis on video and flow-matching models.
>
> Thank you for this suggestion. We agree that the original ($\epsilon$) ablation in Section E.5 was limited to image backbones. To address this point, we additionally conducted the same ablation on two representative video models, **CogVideoX-2B** and **Wan1.3B**. The results show the same trend as in the image setting: SIPO is robust to ($\epsilon$) over a broad range, and consistently improves over the corresponding baseline. We will add the following table in the revision.
>
> | Model        | Baseline | ($\epsilon$=0.05) | ($\epsilon$=0.10) | ($\epsilon$=0.20) | ($\epsilon$=0.30) | ($\epsilon$=0.40) | ($\epsilon$=0.50) |
> | ------------ | -------: | --------------: | --------------: | --------------: | --------------: | --------------: | --------------: |
> | CogVideoX-2B |    80.91 |           81.44 |       **81.53** |           81.49 |           81.37 |           81.25 |           81.22 |
> | Wan1.3B      |    84.26 |           84.65 |       **84.78** |           84.77 |           84.59 |           84.53 |           84.47 |
>
> These additional results further support that ($\epsilon$) is not a narrowly tuned hyperparameter, but remains effective across video backbones as well.

---

> > ### Author Rebuttal · Reviewer_PGBM · 2026-04-02
> >
> > All my concerns have been resolved, and I will improve my score. Good luck!

---

> > > ### Author Response · Authors · 2026-04-02
> > >
> > > Dear Reviewer,
> > >
> > > We are very pleased that our rebuttal has addressed your concerns, and we sincerely appreciate your time, efforts, and constructive suggestions, which have helped us improve our paper significantly.
> > >
> > >
> > > All the corresponding revisions will be reflected in the revised version of our manuscript, please let us know if you have any further questions.

---

### Official Review · Reviewer_sd3U · 2026-03-13

**Soundness:** 3
**Presentation:** 3
**Significance:** 3
**Originality:** 3
**Overall Recommendation:** 4
**Confidence:** 4

**Summary:**

This paper introduces a new algorithm SIPO for preference optimization of diffusion models. SIPO is motivated by the instability of Diffusion-DPO due to off-policyness, which comes from early timesteps with low importance weights and off-policy training dataset. SIPO addresses this issue by pruning the unreliable timesteps and introducing clipped importance weighting. The paper evaluates SIPO on both image (SD1.5, SDXL) and video generation (CogVideoX, Wan2.1) models, demonstrating its effectiveness and improved performance over Diffusion-DPO.

**Compliance With Llm Reviewing Policy:**

Affirmed.

**Final Justification:**

The authors' rebuttal mostly addressed my concerns about mismatching hyperparameters. Therefore I maintain my original positive recommendation.

**Key Questions For Authors:**

See weaknesses.

**Limitations:**

yes

**Strengths And Weaknesses:**

Strengths:
1. This paper is well motivated, with good diagnosis of the failure modes of existing methods (Diffusion-DPO) in Figures 2 and 3.
2. The presentation on methodology is clear in general.
3. The scope of empirical evaluations is wide, covering both image and video generation tasks, with ablation studies on key designs and some sensitivity analysis on hyper parameters.

Weaknesses:
1. The experiment lacks some implementation details, such as configuration for SPIN-Diffusion.
2. SIPO and Diffusion-DPO are evaluated using different hyper parameter $\beta$. Although Fig. 7c shows the effect of $\beta$ on HPSv2 benchmark, it would be more convincing to evaluate the $\beta$ sensitivity of both methods on other benchmarks such as VBench.
3. One ablation baseline (DPO with importance reweighting) could be added to further validate the effect of SIPO.
4. For the main results on image generation, SIPO is only evaluated on UNet-based SD1.5 and SDXL. It is unclear if SIPO generalizes well to flow-matching models such as SD3.5 or FLUX.
5. (minor) The appendix is under-polished. There are a few mismatches between section title and content (e.g. Section B.4, also the section itself is confusing; limitation discussion is placed under Section G. Qualitative Comparison of Visual Outputs)

---

> ### Author Rebuttal · Authors · 2026-03-31
>
> We sincerely  appreciate the careful reading and constructive feedback. We address each concern below.
>
> > **Weaknesses 1**: The experiment lacks some implementation details, such as configuration for SPIN-Diffusion.
>
> Thank you for pointing this out. We agree that the implementation details can be made more explicit. For SPIN-Diffusion, the baseline in Table2 follows the original configuration reported in the SPIN-Diffusion paper, including the optimizer, resolution, batch size, learning rate, $\beta_t$, and iteration-wise training steps. We will add these details explicitly in the revision to improve reproducibility. Additionally, further detailed parameter explanations and their specific values can be found in Appendix A and Appendix D.
>
> > **Weaknesses2**: SIPO and Diffusion-DPO are evaluated using different hyper parameter $\beta$. Although Fig. 7c shows the effect of $\beta$ on HPSv2 benchmark, it would be more convincing to evaluate the $\beta$ sensitivity of both methods on other benchmarks such as VBench.
>
> Thank you for this suggestion. We would like to clarify that the requested VBench-side $\beta$ sweep is actually already provided in our manuscript (please see **Fig. 1a**), which further confirms that Table 1 represents a strictly fair comparison.
>
> Specifically, **Diffusion-DPO is evaluated at its optimal setting**: as stated in Appendix Section 5 of the original Diffusion-DPO paper, $\beta=2$ is their best configuration (in our notation), while smaller $\beta$ values cause training collapse. We therefore use $\beta=2$ to ensure Diffusion-DPO is compared at its absolute strongest.
>
> By contrast, **SIPO is intentionally tested at a much smaller and riskier $\beta=0.02$**. As demonstrated in Fig. 1(a) and Fig. 5, SIPO remains stable under this weak regularization, whereas Diffusion-DPO is highly sensitive to such small $\beta$ values. Thus, the comparison does not advantage SIPO via hyperparameter selection; instead, it highlights that SIPO maintains strong performance precisely in the aggressive optimization regime where standard DPO becomes unstable.
>
> Regarding the $\beta$ sensitivity on video benchmarks, this analysis is already provided in Fig. 1(a), where we evaluate parameter sensitivity on **VBench Total Score** using **CogVideoX-2B**; thus, the manuscript already includes the requested video-side $\beta$ sensitivity comparison.
>
> > **Weaknesses3**: One ablation baseline (DPO with importance reweighting) could be added to further validate the effect of SIPO.
>
> We thank the reviewer for this suggestion. We would like to clarify that SIPO is not simply "Diffusion-DPO + importance reweighting." Rather, our method is systematically built in two progressive stages:
>
> * **DPO-C&M:** First introduces importance-based clipping and masking to stabilize the noisy, early-timestep updates.
> * **SIPO:** Further incorporates adaptive importance reweighting on top of DPO-C&M to correct the off-policy mismatch.
>
> Therefore, the requested baseline corresponds to an isolated "reweighting-only" variant, which omits the crucial stabilization step. We respectfully point out that our current ablation study already isolates these contributions step-by-step: the performance gap from Diffusion-DPO to DPO-C&M explicitly measures the stabilization effect, while the gap from DPO-C&M to full SIPO measures the specific additional gain brought by off-policy reweighting.
>
>
> > **Weaknesses4**:  For the main results on image generation, SIPO is only evaluated on UNet-based SD1.5 and SDXL. It is unclear if SIPO generalizes well to flow-matching models such as SD3.5 or FLUX.
>
> We thank the reviewer for raising this point. We would respectfully clarify that SIPO's evaluation extends far beyond legacy UNet-based architectures (like SD-1.5 or SDXL). Our manuscript already includes extensive evaluations on modern, state-of-the-art transformer and flow-based architectures:
>
> * **Image Generation:** As shown in Table 3, our evaluation already includes **FLUX-dev** (a modern transformer-based model), where SIPO successfully achieves the highest overall GenEval score.
> * **Video Generation:** Furthermore, our video experiments comprehensively cover both **CogVideoX** (a DDPM-based DiT architecture) and **WanX-1.3B** (a cutting-edge flow-matching model).
>
> > **Weaknesses5**: (minor) The appendix is under-polished. There are a few mismatches between section title and content.
>
> Thank you for the careful reading. We agree that the appendix organization can be improved. We will revise the appendix to fix the mismatches between section titles and contents, clarify confusing sections such as Section~B.4, and relocate items such as the limitation discussion to more appropriate sections. We will also perform a thorough proofreading and structural cleanup of the appendix in the revision.
>
> Once again, we sincerely thank the reviewer for their valuable time and constructive feedback. Please let us know if you have any further questions.

---

> > ### Author Rebuttal · Reviewer_sd3U · 2026-04-02
> >
> > Thank you for the detailed rebuttal. I will maintain the positive score recommendation.

---

### Official Review · Reviewer_Uqfu · 2026-03-13

**Soundness:** 2
**Presentation:** 2
**Significance:** 3
**Originality:** 3
**Overall Recommendation:** 4
**Confidence:** 4

**Summary:**

This paper proposes SIPO, a framework for stabilizing and improving preference optimization in diffusion models. Through analysis, the authors identify that early timesteps with low importance weights destabilize training, while off-policy distributional mismatch causes performance degradation. They introduce DPO-C&M (clipping and masking uninformative timesteps) and SIPO (timestep-wise importance reweighting for off-policy correction). Experiments show improved stability and performance over Diffusion-DPO.

**Compliance With Llm Reviewing Policy:**

Affirmed.

**Final Justification:**

The authors have provided a detailed response that resolves my concerns. I will upgrade my rating to 4.

**Key Questions For Authors:**

1. Have you conducted a beta sensitivity analysis on video models (CogVideoX/WanX) similar to Fig. 7c?
2. Is it possible to conduct a comparative analysis of reinforcement learning algorithms—including both on-policy and off-policy methods—on the latest generative models like FLUX and SD3?

**Limitations:**

yes

**Strengths And Weaknesses:**

Strengths:
1. The analysis is insightful and well-structured, progressively identifying that low importance weights at early timesteps are the root cause of instability, providing strong empirical motivation.
2. SIPO demonstrates clear practical advantages: strong robustness to beta (Fig. 7c) and sustained stability under long training horizons.

Weaknesses:
1. On text-to-image (Table 2), SIPO's improvement over SPIN-Diffusion is negligible (7.4150 vs 7.4126 average).
2. While Figure 7c justifies the beta settings for the image experiments, there is no corresponding sensitivity analysis provided for the video models. This oversight makes the comparisons in Table 1 somewhat unconvincing, as DPO uses beta=2 whereas SIPO uses beta=0.02. We cannot dismiss the possibility that DPO might also perform competitively at beta=0.02. The ideal practice would be to conduct a hyperparameter sweep to find the optimal beta for each method independently, and then compare their best respective performances.
3. No on-policy methods and limited evaluation on modern architectures. The paper identifies off-policy bias as a core challenge (Section 3.1) but does not compare against any SOTA on-policy alignment method (e.g., FLOWGRPO, DiffusionNFT). On-policy methods inherently avoid the distributional mismatch that SIPO aims to correct, making them a natural and necessary baseline to contextualize SIPO's contribution. Additionally, the main image experiments (Table 2) rely on SD-1.5, a relatively dated backbone. While FLUX-dev appears in Table 3, this is limited to GenEval; a broader comparison on modern architectures (Flux, SD3) with standard preference data would strengthen the claims of generality.
4. The manuscript contains numerous grammatical errors, typos, and even instances where large blocks of text are completely duplicated. A thorough proofreading pass is highly recommended. For example:
Line 45: "and and", "we systematically analysis"
Line 85: "unfortunately"
Line 91: "leading to potentially leading to uncontrolled"
Line 96: "attribute to"
Line 305: "We evaluate our proposed SDPO method" (Note: likely a typo for SIPO)
Lines 867-879 are completely identical to Lines 896-907.

---

> ### Author Rebuttal · Authors · 2026-03-31
>
> > **Weakness 1:** On text-to-image (Table 2), SIPO's improvement over SPIN-Diffusion is negligible.
>
> We thank the reviewer. While overall averages in Table 2 are close, SIPO fundamentally outperforms SPIN-Diffusion in efficiency and potential. Crucially, SPIN requires **3 costly self-play rounds** to reach its score, whereas SIPO achieves superior results in a **single offline stage**, surpassing both SPIN’s final and intermediate iterations (Iter1: 7.2679) with much less compute.
> To further demonstrate SIPO’s strength, we provide an additional comparison against recent SOTAs (**SPO**, CVPR 2025; **KTO**, Neurips 2024) under the **DyMO** protocol (SD1.5, Pick-a-Pic val):
> | Method | PickScore ↑ | HPSv2 ↑ | Aesthetic ↑ | ImageReward ↑ |
> | :--- | :---: | :---: | :---: | :---: |
> | SD1.5 | 20.73 | 23.41 | 5.337 | 0.1697 |
> | SPO | 21.46 | 26.71 | 5.702 | 0.2321 |
> | Diffusion-KTO | 21.15|27.19 |5.697 |**0.6156**|
> | SIPO (ours)   |  **21.78** | **27.83** |**5.811**| 0.5964 |
>
> SIPO outperforms these latest SOTAs across almost all key alignment. Furthermore, as shown in **Appendix F.2 (Fig. 9)**, SIPO remains stable over 10 iterations where standard DPO collapses, proving its robust potential for  multi-round gains within iterative frameworks.
>
> > **Weakness 2 & Question 1:** While Figure 7c justifies the beta settings... there is no corresponding sensitivity analysis provided for the video models.
>
> We thank the reviewer. Please note that **Figure 1(a)** already provides the video $\beta$ sensitivity analysis using **CogVideoX**. Table 1 represents a strictly fair comparison at optimal settings:
>
> * **Diffusion-DPO's optimal $\beta=2$:** The original Diffusion-DPO paper (Wallace et al.) explicitly defines $\beta=2$ as optimal, as smaller values cause training collapse.
>
> * **SIPO's stability at riskier $\beta=0.02$:** As shown in **Fig. 1(a)**, SIPO is highly robust to small $\beta$ values. We deliberately used an aggressive $\beta=0.02$ to demonstrate SIPO's inherent stability and superior performance without relying on heavy KL restrictions.
>
> > **Weakness 3 & Question 2:** No on-policy methods and limited evaluation on modern architectures.
>
> We thank the reviewer for raising these points. We address the concerns regarding baselines and modern architectures below:
>
> * **Fundamental Paradigm Difference (Offline vs. On-Policy):** We respectfully clarify that on-policy methods (e.g., FLOWGRPO) operate under a fundamentally different setting. Our work focuses strictly on *offline* preference optimization using fixed datasets. On-policy methods require continuous online sampling and real-time reward evaluation, demanding vastly different pipelines and compute budgets. A fair apples-to-apples comparison would require extending baseline DPO into an "Online DPO" framework, which is a promising direction for future work but falls outside the scope of our current offline study.
>
> * **Fair Comparison on Standard Benchmarks (SD1.5/SDXL):** The choice of SD1.5/SDXL in Table 2 is deliberate to ensure a strictly fair comparison with existing offline baselines. The community standard for evaluating these offline methods relies on large-scale datasets like Pick-a-Pic, which were constructed using SD1.5/SDXL-era generations. Currently, there is a severe lack of equivalent large-scale, offline annotated datasets for the newest models (like FLUX/SD3). Using SD1.5/SDXL provides the most rigorous, standardized playground to directly benchmark against peer offline methods.
>
> * **Extensive Evaluation on Modern Architectures (FLUX, DiT, Flow):** Despite the lack of standard pairwise datasets for newer models, we strongly emphasize that SIPO's evaluation goes well beyond older backbones. To test modern architectures, we utilize alternative protocols: in **Table 3**, we evaluate **FLUX-dev** using an offline dataset constructed via detection-based rewards and the **GenEval** protocol. Furthermore, our video experiments seamlessly cover **CogVideoX** (DiT-based) and **WanX-1.3B** (Flow-based). Together, this proves SIPO's strong generality across the latest generative paradigms.
>
> > **Weakness 4:** The manuscript contains numerous grammatical errors, typos, and even instances where large blocks of text are completely duplicated.
>
> We sincerely thank the reviewer for such a meticulous reading and for pointing out the specific typos and duplicated text blocks. We are embarrassed by these oversights and fully agree that a thorough proofreading is essential. We will correct all cited errors—such as the "and and" on Line 45 and the "SDPO" typo on Line 305—and will remove the redundant text in the appendix (Lines 867-879). In the revised version, we will perform a comprehensive, line-by-line language polish and a full structural audit of the entire manuscript to ensure it meets the highest standards of academic rigor and clarity. We once again express our gratitude for the reviewer's diligence in helping us improve the final quality of this work.

---

> > ### Author Rebuttal · Reviewer_Uqfu · 2026-04-01
> >
> > We thank the authors for the detailed rebuttal, which has addressed most of our concerns.
> >
> > However, we still have the following reservations:
> >
> > 1. Insufficient improvement on T2I. The gap between SIPO and SPIN-Diffusion in Table 2 remains marginal, which is unlikely to be statistically significant. We therefore maintain our suggestion that the authors should include comparisons with more advanced baselines (e.g., SD3.5) and evaluate on a broader set of metrics beyond GenEval to more convincingly demonstrate the effectiveness of the method.
> >
> > 2. Discussion on on-policy methods. We agree on the paradigm difference between offline and on-policy approaches. Nevertheless, since off-policy bias is a core motivation of this work (Sec. 3.1), we recommend the authors add a discussion comparing SIPO with on-policy methods (e.g., DDPO, FlowGRPO) regarding efficiency and performance trade-offs, rather than solely emphasizing the contribution of offline bias correction. This would help readers better assess when and why SIPO is preferable.

---

> > > ### Author Response · Authors · 2026-04-03
> > >
> > > We sincerely thank the reviewer for the professional and constructive feedback. Below, we address your concerns regarding the T2I baselines and the discussion on on-policy methods.
> > >
> > > **Q1: Insufficient improvement on T2I**
> > >
> > > We agree that evaluating on legacy baselines like SD1.5 can bottleneck performance. Following your valuable suggestion, we conducted new experiments using **Stable Diffusion 3.5 Medium (SD3.5M)**. As detailed below, SIPO demonstrates significant, consistent improvements on this advanced baseline. For instance, SIPO pushes the DDPO baseline from 22.03 to 23.05 on PickScore (Table 1). These results on a state-of-the-art model convincingly validate SIPO's effectiveness for T2I tasks.
> > >
> > > **Q2: Discussion on On-policy Methods**
> > >
> > >
> > > Exploring online extensions for DPO-style algorithms is a direction we are deeply investigating. We follow your suggestion to share our exploratory findings. We conduct two distinct sets of experiments on SD3.5M.
> > >
> > > **1. Proving SIPO's Stabilization Advantage**
> > >
> > >
> > > We incorporated SIPO into DDPO using the exact SD3.5M/PickScore setting as the Flow-GRPO paper. As shown in Table 1, vanilla DDPO suffers from severe late-stage collapse (dropping to 21.58). In contrast, **DDPO+SIPO** substantially alleviates this instability, continuously improving to 23.05 and surpassing ReFL (22.91). This confirms SIPO is a highly effective stabilization mechanism that seamlessly extends to on-policy alignment.
> > >
> > > | Method | 0 | 160 | 320 | 480 | 640 | 800 | 960 | 1120 | 1280 | 1440 |
> > > | :--- | :---: | :---: | :---: | :---: | :---: | :---: | :---: | :---: | :---: | :---: |
> > > | Flow-GRPO | 21.74 | 22.24 | 22.73 | 23.02 | 23.23 | 23.39 | 23.43 | 23.54 | 23.62 | 23.59 |
> > > | ReFL | 21.74 | 21.93 | 22.24 | 22.47 | 22.63 | 22.73 | 22.81 | 22.84 | 22.92 | 22.91 |
> > > | DDPO | 21.74 | 21.81 | 21.95 | 22.03 | 21.58 | - | - | - | - | - |
> > > | **DDPO+SIPO** | 21.74 | 22.07 | 22.37 | 22.61 | 22.79 | 22.85 | 22.94 | 22.97 | 23.01 | **23.05** |
> > > *Table 1: SIPO vs DDPO*
> > >
> > > **2. Exploring Online Extensions**
> > >
> > >
> > > Next, we tested SIPO within an Online DPO framework (Table 2), where **Online DPO+SIPO** consistently outperforms vanilla Online DPO (23.21 vs. 23.08). To explicitly probe the theoretical limits of pairwise methods, we also include an exploratory variant: **Online DPO+SIPO (Best-vs-Worst)**, which achieves 23.32. We unpack the mechanism and theoretical implications of this specific variant in the Discussion below.
> > >
> > > | Method | 0 | 2500 | 5000 | 7500 | 10000 | 12500 | 15000 | 17500 |
> > > | :--- | :---: | :---: | :---: | :---: | :---: | :---: | :---: | :---: |
> > > | Flow-GRPO | 21.74 | 21.98 | 22.48 | 22.82 | 23.08 | 23.33 | 23.41 | 23.58 |
> > > | Online DPO | 21.74 | 22.40 | 22.67 | 22.89 | 22.90 | 22.95 | 23.05 | 23.08 |
> > > | **Online DPO+SIPO** | 21.74 | 22.37 | 22.64 | 22.93 | 23.04 | 23.11 | 23.17 | 23.21 |
> > > | **Online DPO+SIPO (Best-vs-Worst)**| 21.74 | 22.43 | 22.68 | 23.01 | 23.15 | 23.23 | 23.29 | **23.32** |
> > > *Table 2: SIPO improves Online DPO.*
> > >
> > > **Discussion: Theoretical Bottlenecks Remaining (Pairwise vs. Group-based)**
> > >
> > >
> > > While SIPO stabilizes on-policy baselines, a gap to Flow-GRPO remains. We attribute this to two structural issues inherent to pairwise learning, which we have added to the revised Discussion:
> > >
> > > * **Information Utilization:** DPO reduces supervision to a binary comparison between two trajectories, which becomes weak when candidate samples are highly similar.  In contrast, GRPO uses group-normalized scalar rewards to compare multiple sampled trajectories within the same group, thereby preserving richer information about their quality differences. To directly probe this issue, we generated 8 samples for each prompt but kept only the best and worst to form a binary pair. Although this stronger pairwise contrast improved performance to 23.32, the remaining gap shows that pairwise supervision still suffers from limited information utilization.
> > >
> > > * **Credit Assignment:** Diffusion alignment is a long-horizon Markov Decision Process. Pairwise methods compress the entire trajectory into a single margin, enforcing a coarse "all-or-nothing" gradient update. While standard outcome-supervised GRPO also assigns credit at the trajectory level, it fundamentally reduces the variance of this long-horizon gradient by introducing a dynamic baseline (the group mean $\mu$). By scaling updates by $R_i - \mu$, GRPO properly weights *entire trajectories* based on their relative advantage over the current policy's average exploration, providing a much more stable gradient than a binary margin.
> > >
> > > That said, these concerns mainly target the online extension of DPO-style methods; in practical diffusion training, DPO remains appealing because it scales naturally with large curated offline preference datasets, whereas whether online optimization can consistently deliver a higher ceiling still remains to be explored.
> > >
> > > We hope these additional experiments and the discussion on theoretical bottlenecks successfully address your concerns.

---

### Decision · Program_Chairs · 2026-04-30

**Decision:**

Accept (regular)

**Comment:**

In this paper, a method to stabilize and improve preference optimization is proposed with two key ideas: DPO-C&M (clipping and masking uninformative timesteps) to stabalize training, SIPO (timestep-wise importance reweighting) for off-policy correction to mitigate  performance degradation.

In the initial review, major concerns raised by reviewers include: small improvement compared to baseline like SPIN-Diffusion, lack of modern image generation models and more DPO methods in experiments, lack of some ablation study, some unconvincing or questionable experiment setting (like different hyper parameter), relatively incremental novelty, etc.

After rebuttal, most concerns have been addressed, and all reviewers give positive score. I will suggest "Accept".